# A comparison of risk factors for cryptosporidiosis and non-cryptosporidiosis diarrhoea: A case-case-control study in Ethiopian children

**Øystein Haarklau Johansen** [1,2]*, **Alemseged Abdissa**[3,4], **Mike Zangenberg**[5,6], **Zeleke Mekonnen**[3], **Beza Eshetu**[7], **Bizuwarek Sharew**[3], **Sabrina Moyo**[1], **Halvor Sommerfelt** [8,9], **Nina Langeland** [1,10], **Lucy J. Robertson** [11], **Kurt Hanevik** [1,10]

1 Department of Clinical Science, University of Bergen, Bergen, Norway, 2 Department of Microbiology, Vestfold Hospital Trust, Tønsberg, Norway, 3 School of Medical Laboratory Sciences, Jimma University, Jimma, Ethiopia, 4 Armauer Hansen Research Institute, Addis Ababa, Ethiopia, 5 Department of Infectious Diseases, Copenhagen University Hospital, Hvidovre, Denmark, 6 Department of Immunology and Microbiology, Centre for Medical Parasitology, University of Copenhagen, Copenhagen, Denmark, 7 Department of Paediatrics, Jimma Medical Centre, Jimma University, Jimma, Ethiopia, 8 Centre for Intervention Science in Maternal and Child Health, Centre for International Health, University of Bergen, Bergen, Norway, 9 Cluster for Global Health, Division for Health Services, Norwegian Institute of Public Health, Oslo, Norway, 10 Norwegian National Advisory Unit on Tropical Infectious Diseases, Department of Medicine, Haukeland University Hospital, Bergen, Norway, 11 Parasitology, Department of Paraclinical Sciences, Faculty of Veterinary Medicine, Norwegian University of Life Sciences, Ås, Norway

* haarklau@gmail.com

**Data Availability Statement:** All relevant data are within the manuscript and its Supporting Information files.

## Abstract

### Background

Cryptosporidiosis is a major cause of diarrhoea in young children in low-and-middle-income countries. New interventions should be informed by evidence pertaining to risk factors and their relative importance. Inconsistencies in the literature may to some extent be explained by choice of methodology, furthermore, most previous risk factor studies compared cryptosporidiosis cases to diarrhoea cases of other aetiologies rather than with controls without diarrhoea.

### Methodology/Principal findings

We investigated a broad set of factors in under-2-year-olds presenting with diarrhoea to a hospital and a health center in southwestern Ethiopia. We applied quantitative cut-offs to distinguish between cryptosporidiosis and incidental *Cryptosporidium* infection or carriage, a hierarchical causal framework to minimize confounding and overadjustment, and a case-case-control design, to describe risk factors for both cryptosporidiosis and non-cryptosporidiosis diarrhoea. Moderate and severe acute malnutrition were strongly associated with both cryptosporidiosis and non-cryptosporidiosis diarrhoea. Previous healthcare attendance and low maternal education were only associated with cryptosporidiosis, whereas unsafe child stool disposal, prematurity and early cessation of exclusive breastfeeding were significantly associated with non-cryptosporidiosis diarrhoea only. By estimation of population

**Funding:** This work was supported by the Research Council of Norway [ grant number 255571, to N.L.; grant number 223269 to H.S.; https://www.forskningsradet.no/en/ ]; Bill and Melinda Gates Foundation [ grant number OPP1153139, to N.L.; https://www.gatesfoundation.org/ ]; Norwegian Society for Medical Microbiology [ to Ø.H.J.; https://www.legeforeningen.no/mikrobio ]; University of Bergen [ to Ø.H.J.; https://www.uib.no/en ] and Vestfold Hospital Trust [ to Ø.H.J.; https://www.siv.no/ ]. The funders had no role in study design, data collection and analysis, decision to publish, or preparation of the manuscript.

**Competing interests:** The authors have declared that no competing interests exist.

attributable fractions, socioeconomic factors—specifically low maternal education—and public tap water use, were apparently more important risk factors for cryptosporidiosis than for non-cryptosporidiosis diarrhoea.

## Conclusions/Significance

Nutritional management of moderate acute malnutrition may be an effective intervention against cryptosporidiosis, particularly if combined with targeted therapy for cryptosporidiosis which, again, may mitigate nutritional insult. Focused caregiver education in healthcare settings and follow-up of children with acute malnutrition may prevent or improve outcomes of future episodes of cryptosporidiosis.

### Author summary

There are puzzling contradictions between reported risk factors for cryptosporidiosis and for paediatric diarrhoea in general. We suspected that these differences are more related to different methodological approaches rather than real differences in the underlying epidemiology. To address this, we applied several epidemiological tools that, to our knowledge, have not previously been combined in a risk-factor analysis for cryptosporidiosis: 1) an underlying conceptual framework 2) the use of pragmatic "real-life" inclusion criteria, 3) quantitative cutoffs for case ascertainment, and 4) a case-case-control design, to allow side-by-side comparison of risk factors for cryptosporidiosis and non-cryptosporidiosis diarrhoea. Caregiver-related socioeconomic factors, public-tap water use, previous healthcare attendance, and moderate acute malnutrition were apparently more strongly associated with cryptosporidiosis than with non-cryptosporidiosis diarrhoea. Our results suggest giving priority to nutritional management of moderate acute malnutrition and exploring healthcare-initiated interventions, with focused caregiver education and closer follow-up of children who present for healthcare or have acute malnutrition, in order to prevent or ameliorate outcomes of future episodes of cryptosporidiosis.

## Introduction

Infectious diarrhoea is an important cause of death in young children [1–3]. In children under 2 years of age presenting with diarrhoea, important predictors of death within 2 months include acute malnutrition [4], infection with enteropathogenic *E coli* or enterotoxigenic *E coli* expressing the human variant of the thermostable toxin, and infection with *Cryptosporidium* [2]. Excess mortality after cryptosporidiosis in infancy has also been reported [5].

Interventions that target well-established risk factors are more likely to be effective. However, there are evidence-gaps and unresolved discrepancies in the literature between risk factors for cryptosporidiosis and risk factors for pediatric diarrhoea in general [6, 7], both for environmental factors [8, 9], animal exposures [10, 11], breastfeeding and malnutrition [8, 9]. The comparators used in most risk factor studies are diarrhoea cases without *Cryptosporidium* infection, i.e., case-case comparisons [12–19]. We have not identified any case-control studies that distinguished between *Cryptosporidium* infection and cryptosporidiosis, i.e., diarrhoea attributed to *Cryptosporidium* infection [20, 21] compared with children with no diarrhoea, that investigated hand hygiene, perinatal factors, and acute malnutrition in the same analysis

[4, 11], or that adequately addressed confounding and the internal relationship between risk factors using a pre-defined conceptual causal framework.

The aim of our study was to identify and compare a broad range of risk factors for cryptosporidiosis using community controls without diarrhoea. We included several measures of socioeconomic status, drinking water source, sanitation standards, perinatal factors, factors related to caregiver hygiene, and previous illness. We also investigated the association between severe and moderate acute malnutrition (SAM or MAM) and healthcare-presenting cryptosporidiosis [4]. In order to minimize the risk of either overestimating or underestimating the relative importance of any of the above factors, we used a predefined causal conceptual framework in the analysis, primarily to guide decisions about confounder adjustment. Furthermore, in order to distinguish between those risk factors that are unique to cryptosporidiosis, and those that are common with non-cryptosporidiosis diarrhoea, comparisons of cryptosporidiosis versus non-diarrhoea controls are presented side-by-side with comparisons of non-cryptosporidiosis diarrhoea versus non-diarrhoea controls (a case-case-control design).

## Methods

### Ethics statement

Jimma University IRB (Reference: RPGC/610/2016), the Ethiopian National Research Ethics Review Committee (Reference: JU JURPGD/839/2017) and the Regional Committee for Medical and Health Research Ethics of Western Norway (Reference: 2016/1096) approved the study. Formal written consent was obtained from the children's parents or guardians.

### Study design

Whereas community studies with longitudinal stool sampling are more appropriate for identifying risk factors for asymptomatic or less-severe infection [9, 22], we deliberately focused on risk factors in children who sought care for diarrhoea. The reason was twofold: clinical presentation is likely to be a proxy for severity [2], and low-resource healthcare centres and hospitals might represent under-utilized opportunities for simple, low-cost interventions against cryptosporidiosis that could have high impact.

To minimize the risk of confounding and overadjustment, while maintaining a pragmatic focus on cryptosporidiosis, we used three epidemiological tools that are well established, but that have not, to our knowledge, been previously combined in a risk-factor analysis for childhood diarrhoea: 1) hierarchical conceptual frameworks, where risk factors are organized in levels of a hierarchy [23, 24], 2) improved case ascertainment, using a reference standard that includes quantitative cutoffs, allowing us to distinguish between incidental infection and cryptosporidiosis [20, 21, 25, 26], and 3) a case-case-control design, an approach originally developed to study risk factors for multidrug-resistant bacterial infections [27, 28], in order to distinguish between those risk factors unique to cryptosporidiosis, and those common with non-cryptosporidiosis diarrhoea.

### Selection of cases and controls

This analysis used data from a case-control study nested within the CRYPTO-POC study, a diagnostic accuracy study of light-emitting diode auramine-phenol staining microscopy and a rapid antigen test strip for near-patient diagnosis of cryptosporidiosis, conducted in southwest Ethiopia from December 2016 to July 2018 [26, 29]. In brief, the study enrolled children younger than 5 years who presented to Jimma Medical Center (JMC; formerly Jimma University Specialized Hospital) or Serbo Health Centre (SHC; approximately 16 km from JMC) with

diarrhoea (three or more loose stools within the previous 24 hours), or dysentery (at least one loose stool with stains of blood within the previous 24 h), and who lived within two nearby predefined geographical catchment areas. There was no exclusion of cases with prolonged (7–13 days) or persistent (≥14 days) diarrhoea. Community controls without diarrhoea (in the preceding 48 hours) were enrolled concurrently by weekly recruitment plans, using frequency matching to cases by age stratum (0–5 months, 6–11 months, 12–23 months) and geographical location of households (S1 Appendix). The sample size was determined by the parent study (S1 Appendix). Stool testing of controls for asymptomatic *Cryptosporidium* infection is not part of the current analysis, but was conducted as part of the CRYPTO-POC study [26].

## Data collection

Study nurses obtained informed written consent from the children's caregivers, collected demographic, exposure, and clinical data using standardized questionnaires, tested the children for HIV, and asked cases to provide a stool sample [26]. Stools were tested for *Cryptosporidium* by a composite reference standard comprising antigen detection by ELISA, oocyst detection and quantification by immunofluorescent antibody test microscopy (qIFAT), and DNA quantification by qPCR. We used our previously established quantitative qIFAT and qPCR cutoffs (725 oocysts/gram and 231302 copies/gram, respectively) for diarrhoea-associated infection; these cutoffs were applied without analysis for other possible co-infecting diarrhoeal pathogens [26]. The composite reference standard was considered positive if two or more reference tests were positive (and greater than the quantitative cutoff) and negative if two or more reference tests were negative (or less than the quantitative cutoff). A diarrhoea case was defined as cryptosporidiosis if the composite reference standard was positive and as non-cryptosporidiosis diarrhoea (NCrD) if negative [26]. The included variables were obtained from key published papers on risk factors for *Cryptosporidium* infection and/or cryptosporidiosis, and other risk factors known to be important for diarrhoea in general [6]. Acute malnutrition (MAM or SAM) was defined using mid-upper arm circumference (MUAC) thresholds in 6-59-month-olds, and by WHO weight-for-height z-scores in children < 6 months [30], and/or presence of bilateral oedema involving at least the feet; see S1 Appendix for these and other variable definitions.

## Statistical methods

Double data entry was done with EpiData (version 3.1). All data were analysed in R (version 4.0.3) and RStudio (version 1.3). Missing values for exposure variables were multiply imputed (100 imputations) by chained random forests (500 trees) with predictive mean matching, using the R package missRanger (v.2.1). Odds ratios (OR) and their 95% confidence intervals (CI) were estimated by unconditional mixed model logistic regression, using the R package lme4 (v.1.1). We used a case-case-control design [27, 28], where the cryptosporidiosis and NCrD case sets were compared to the same non-diarrhoea control group, with results presented side-by-side for comparison.

The multivariable analysis followed a step-by-step approach using a hierarchical conceptual framework, as first outlined by Victora *et al* [23], a method that enables adjustment for risk factors that are causally distal to disease, while, at the same time, avoiding the common mistake of underestimating distal risk factors by adjusting for more proximal ones that act as mediators of their effect [31, 32]. To accomplish this, the analysis was governed by a predefined conceptual model [23] for the causal and hierarchical relationships between the proposed risk factors (Fig 1).

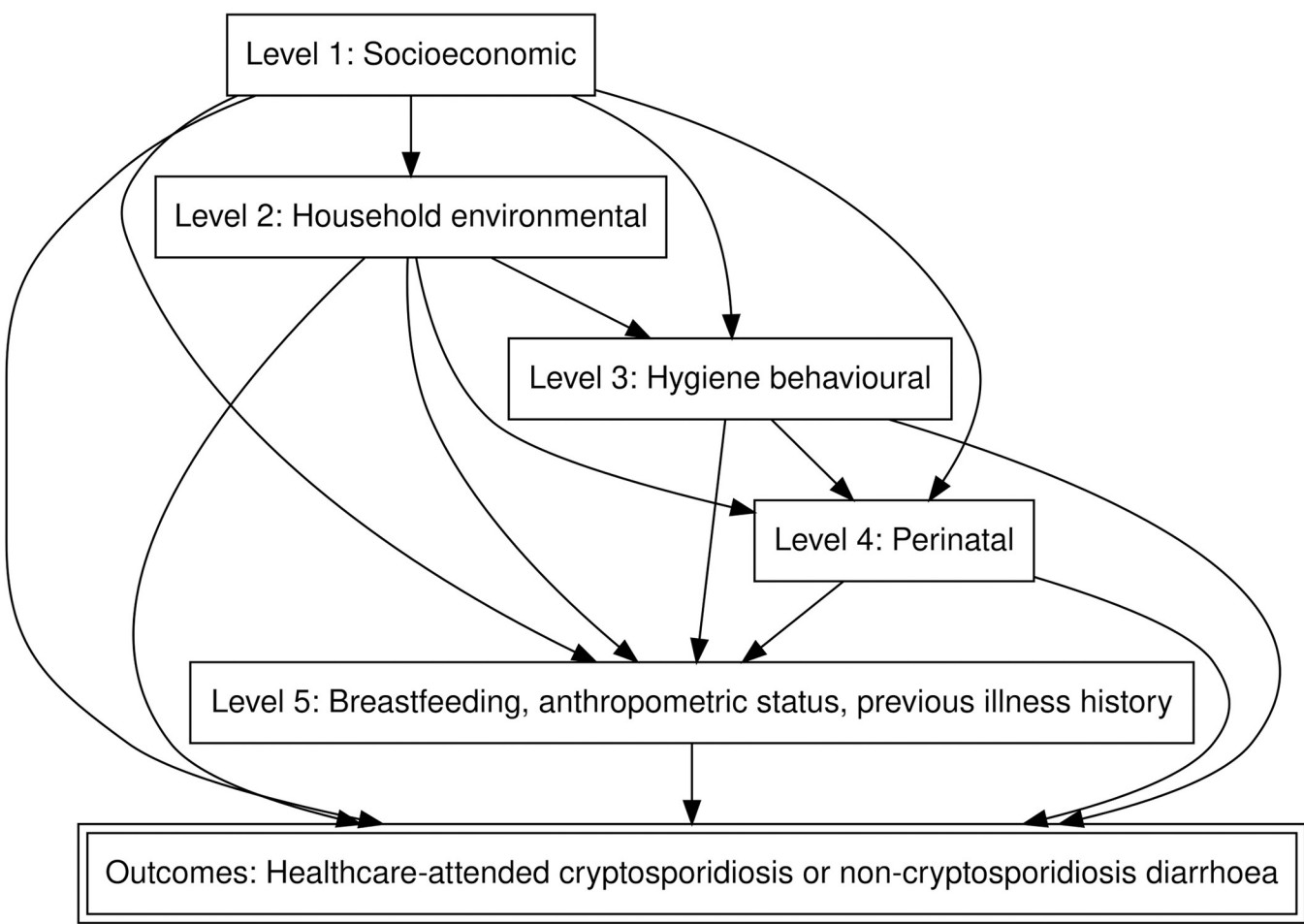

**Fig 1. Hierarchical conceptual framework for the relationship between the putative risk factors for cryptosporidiosis and non-cryptosporidiosis diarrhoea.**

The conceptual framework defines five hierarchical risk-factor levels [23, 24], where socio-economic factors (level 1) are considered most distal to the outcomes of cryptosporidiosis and NCrD, and where nutritional factors and factors related to previous illness (level 5) are most proximal. Any level can be caused, in part or in full, by levels more distal to it, and the outcomes can be caused both directly and indirectly by factors at all levels. All models included adjustment for age (in months), gender, study site, and enrolment season (divided into six three-month intervals), i.e., the base adjustment set. Initial base-adjusted models were followed by intra-level models before undertaking a full hierarchical analysis. Importantly, the final estimate for the overall effect of a distal variable was the estimate derived before the introduction of more proximal-level risk factors. See S1 Appendix for details on the step-by-step modelling strategy.

Population attributable fractions (PAF) were estimated separately for cryptosporidiosis and NCrD for all risk factors that were significantly associated (i.e. with a p-value $<0.05$) in the hierarchical analysis, with the formula $PAF = Prevalence \ x \left(1 - \frac{1}{OR}\right)$, using the imputed prevalence of the risk factor in the case group and the OR estimate for the association between the risk factor and either disease [33]. Summary PAFs were calculated for each risk-factor level, derived from models that were adjusted for more distal levels, but not including more

proximal levels [24]. A summary PAF for all levels was also calculated, by taking the complement of the PAF at each level. A separate analysis was conducted to estimate the fraction of PAF that could potentially be explained by mediation through more proximal levels, using a variant of the traditional "difference method" of quantifying mediation, which is based on comparing odds ratios between two logistic regression models, where one model includes adjustment for a mediator, and the other model does not (S1 Appendix) [34].

## Results

Diarrhoeal disease age distribution differed by case set; 95% (59/62) of cryptosporidiosis cases were younger than 24 months compared with 71% (432/607) of NCrD cases (Fig 2). As our priority was cryptosporidiosis risk factors, and to minimize the risk of residual confounding by age, we limited the statistical analysis to 0-23-month-olds. This included 1216 children aged 0–23 months, of whom 59 were cases with cryptosporidiosis, 432 cases with NCrD, and 725 controls. For details of screening, eligibility, and inclusion, see previous works [26, 29] and the study flowchart (Fig 2).

Table 1 shows the distribution of demographic characteristics, and distribution by enrolment period and study site, in the case sets and in the controls. Cryptosporidiosis numbers were not evenly distributed throughout the study period, with most cases during the late dry season (Feb-Apr) and early wet season months (May-Jul). A similar pattern was not seen for NCrD.

We first compared exposures between controls and each of the two case sets. There were no missing data in the base adjustment set, and no exposure variable had over 3.4% missing values (Table A in S1 Appendix). Risk factor associations are presented according to the structure of the statistical analysis; first, exposure-outcome associations estimated by univariable models (Table 2); second, risk-factor exposures, while considering other exposures at the same hierarchical level, estimated by "intra-level" multivariable models (Table 3), starting with the most distal (i.e., level 1) risk factors, and ending with the most proximal risk factors (level 5). Finally, risk-factor associations that take all hierarchical levels into account are presented, as estimated from the step-by-step hierarchical analysis (Table 4).

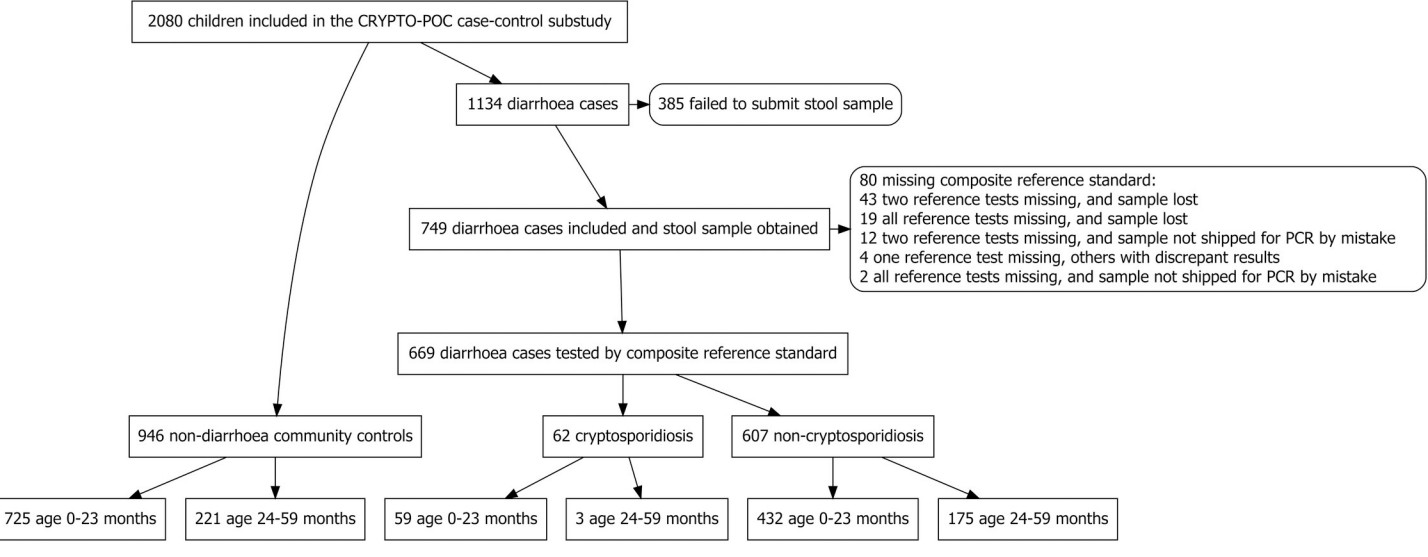

**Fig 2. Study flowchart.** For details on screening and eligibility, see flow diagram in previous publication [29]).

**Table 1. Distribution of diarrhoea cases with and without cryptosporidiosis and non-diarrhoea controls, according to gender and a priori defined confounding demographic variables.**

| Characteristic | Controls; n (%) (N = 725) | Cryptosporidiosis; n (%) (N = 59) | Non-cryptosporidiosis; n (%) (N = 432) |
|---|---|---|---|
| | | **Diarrhoea cases** | |
| **Age, in months** | | | |
| < 6 | 92(13) | 3(5) | 54(12) |
| 6–11 | 322(44) | 34(58) | 189(44) |
| 12–23 | 311(43) | 22(37) | 189(44) |
| **Gender[a]** | | | |
| Female | 331(46) | 31(53) | 170(39) |
| Male | 394(54) | 28(47) | 262(61) |
| **Study site** | | | |
| Jimma hospital | 332(46) | 36(61) | 186(43) |
| Serbo health center | 393(54) | 23(39) | 246(57) |
| **Season** | | | |
| Feb–Apr: dry season | 52(7) | 18(31) | 52(12) |
| May–Jul: wet season | 120(17) | 18(31) | 102(24) |
| Aug–Oct: wet season | 142(20) | 4(7) | 84(19) |
| Nov–Jan: dry season | 194(27) | 4(7) | 101(23) |
| Feb–Apr: dry season (second enrolment year) | 132(18) | 10(17) | 60(14) |
| May–Jul: wet season (second enrolment year) | 85(12) | 5(8) | 33(8) |

[a] Gender: evidence of selection bias due to missing outcome in some cases (S1 Appendix).

Table 2 shows the corresponding ORs for the univariable relationships, with estimates for cryptosporidiosis and NCrD side-by-side and grouped by risk-factor level. These models contained only the base adjustment set in addition to the exposure and outcome variables, i.e., adjustment for age, gender, site, and season. Of the socioeconomic (level 1) variables, having a primary caregiver other than the mother was a strong risk factor for diarrhoea, and tended to be more strongly associated with cryptosporidiosis than with NCrD. Socioeconomic factors appeared to affect risk differently for cryptosporidiosis and NCrD; household lacking ownership of key assets was only associated with NCrD, whereas low maternal education was a risk factor specifically for cryptosporidiosis. For NCrD, there was a borderline trend of increased risk by household size. After the univariable models, the next step was adjustment for the other level-1 risk factors, but this had little impact on the observed associations (Table 3).

The following household environmental (level 2) variables were associated with NCrD in the univariable models, ordered by strength of the association from high to low: primary water source from a public piped water tap or from an unimproved source, household pre-treatment of drinking water, and cattle ownership. Of these, collecting drinking water from a public tap was also associated with cryptosporidiosis. Interestingly, however, consumption of surface water or water from unimproved sources, was not. Unimproved sanitation was neither associated with cryptosporidiosis nor NCrD, although this estimate had a wide margin of uncertainty due to evidence of differential exposure misclassification (S1 Appendix). All risk factors from the base-adjusted models remained statistically significant in the intra-level analysis (Table 3), except for the cattle ownership association with NCrD (OR 1.2, 95% CI 0.8 to 1.7).

Several hygiene-related (level 3) factors and perinatal (level 4) factors were associated with NCrD in the univariable models: handwashing without soap, unsafe disposal of child stool, premature delivery, and delivery by caesarean section (Table 2). All remained significant in the

**Table 2. Risk factors for cryptosporidiosis diarrhoea and non-cryptosporidiosis diarrhoea in children under 2 years old, compared with non-diarrhoea controls; odds ratios estimated from univariable models[a].**

| Characteristic | Reference level | Controls[b] (%) (N = 725) | Diarrhoea cases Cryptosporidiosis[b] (%) (N = 59) | Non-cryptosporidiosis[b] (%) (N = 432) | Cryptosporidiosis diarrhoea vs controls OR | 95% CI for OR from | 95% CI for OR to | Linear trend $P$ | Non-cryptosporidiosis diarrhoea vs controls OR | 95% CI for OR from | 95% CI for OR to | Linear trend $P$ |
|---|---|---|---|---|---|---|---|---|---|---|---|---|
| **Level 1 – Socioeconomic factors** | | | | | | | | | | | | |
| Maternal education | | | | | | | | 0·02[c] | | | | 0·69[c] |
| < 1 year | ≥ 8 years | 27·4 | 35·6 | 29·9 | 2·4 | 1·1 | 4·9 | | 1·1 | 0·78 | 1·4 | |
| 1–7 years | ≥ 8 years | 37·7 | 32·2 | 34·7 | 1·2[d] | 0·58 | 2·3 | | 0·93[d] | 0·70 | 1·2 | |
| Primary caregiver is not the child's mother | Mother primary caregiver | 1·8 | 10·2 | 5·8 | 6·3 | 2·0 | 20·0 | | 3·5 | 1·7 | 7·0 | |
| Number of key assets owned by the household ≤ 2 | 3–7 key assets owned | 16·0 | 13·6 | 24·1 | 0·93 | 0·42 | 2·1 | 0·10[e] | 1·7 | 1·3 | 2·3 | 0·02[e] |
| Number of household members | | | | | | | | 0·27[e] | | | | 0·06[e] |
| 4–5 | < 4 members | 46·6 | 47·5 | 39·6 | 0·72 | 0·38 | 1·4 | | 0·88 | 0·65 | 1·2 | |
| ≥ 6 | < 4 members | 27·7 | 22·0 | 36·1 | 0·74 | 0·34 | 1·6 | | 1·4 | 0·97 | 1·9 | |
| **Level 2 –Household environmental factors** | | | | | | | | | | | | |
| Persons per room ≥ 2 | < 2 per room | 93·5 | 86·4 | 90·7 | 0·52 | 0·22 | 1·2 | | 0·72 | 0·46 | 1·1 | |
| Animals owned by the household | | | | | | | | | | | | |
| Cattle | No cattle | 33·7 | 27·1 | 41·7 | 1·1 | 0·50 | 2·6 | | 1·4 | 1·1 | 1·7 | |
| Chickens | No chickens | 32·1 | 27·1 | 33·6 | 1·1 | 0·55 | 2·0 | | 1·0 | 0·81 | 1·4 | |
| Dogs | No dogs | 8·6 | 13·6 | 6·9 | 1·8 | 0·80 | 4·3 | | 0·76 | 0·48 | 1·2 | |
| Goats | No goats | 7·4 | 6·8 | 9·5 | ND | ND | ND | | 1·3 | 0·83 | 2·0 | |
| Horses, donkeys, or mules | No horses, donkeys, or mules | 9·1 | 6·8 | 12·5 | ND | ND | ND | | 1·4 | 0·92 | 2·0 | |
| Sheep | No sheep | 14·8 | 10·2 | 14·1 | 1·2 | 0·45 | 3·1 | | 0·99 | 0·70 | 1·4 | |
| Other | No other animals | 3·2 | 1·7 | 1·4 | ND | ND | ND | | 0·42 | 0·17 | 1·1 | |
| Any even-toed ungulate | No even-toed ungulates | 35·9 | 32·2 | 42·6 | 1·6 | 0·69 | 3·6 | | 1·3 | 0·99 | 1·6 | |
| Any animal | No animals | 49·2 | 45·8 | 50·2 | 1·1 | 0·61 | 2·1 | | 0·99 | 0·77 | 1·3 | |
| Sanitation facility[f] | | | | | | | | 0·34[c] | | | | 0·09[c] |
| Improved, but shared | Unimproved facility | 8·6 | 11·9 | 19·0 | 0·63 | 0·15 | 2·7 | | 1·3 | 0·68 | 2·3 | |
| Improved, and not shared | Unimproved facility | 8·1 | 37·3 | 28·7 | 1·8 | 0·55 | 5·6 | | 1·8 | 0·92 | 3·4 | |
| Access to "improved sanitation" (by the WHO definition)[f] | Unimproved or shared facility | 8·1 | 37·3 | 28·7 | 2·1 | 0·72 | 5·9 | | 1·6 | 0·88 | 2·9 | |
| Water source for the household | | | | | | | | 0·98[c] | | | | <0·01[c] |
| Public tap | Private tap | 13·7 | 37·3 | 24·5 | 3·7[d] | 1·8 | 7·3 | | 2·8[d] | 1·9 | 4·0 | |
| Surface or rainwater, unprotected well, borehole, or protected spring | Private tap | 30·6 | 15·3 | 35·9 | 0·99[d] | 0·41 | 2·4 | | 1·9[d] | 1·4 | 2·8 | |

*(Continued)*

**Table 2.** (*Continued*)

| Characteristic | Reference level | Controls[b] (%) (N = 725) | Cryptosporidiosis[b] (%) (N = 59) | Non-cryptosporidiosis[b] (%) (N = 432) | Cryptosporidiosis diarrhoea vs controls OR | 95% CI for OR from | to | Linear trend P | Non-cryptosporidiosis diarrhoea vs controls OR | 95% CI for OR from | to | Linear trend P |
|---|---|---|---|---|---|---|---|---|---|---|---|---|
| Water treated by the household (chemicals, boiling or filtering) before drinking | No water treatment | 5·2 | 5·1 | 9·5 | ND | ND | ND | | 1·9 | 1·2 | 3·1 | |
| **Level 3 –Hygiene behaviour** | | | | | | | | | | | | |
| Last stool disposal (from any of the caregiver's children) "unsafe" by the WHO definition | Safe disposal (i.e., in toilet/latrine, or buried) | 55·7 | 66·1 | 75·7 | 1·50 | 0·77 | 3·0 | | 2·4 | 1·7 | 3·3 | |
| Caregiver will normally wash hands | | | | | | | | | | | | |
| before meals | not before meals | 94·6 | 93·2 | 92·1 | ND | ND | ND | | 0·62 | 0·38 | 1·0 | |
| before preparing food for the child | not before preparing food for the child | 74·2 | 78·0 | 72·9 | 1·0 | 0·53 | 2·0 | | 0·88 | 0·67 | 1·2 | |
| after a toilet visit | not after a toilet visit | 68·6 | 64·4 | 70·1 | 0·93 | 0·52 | 1·7 | | 1·1 | 0·85 | 1·5 | |
| without soap | with soap | 3·3 | 5·1 | 6·9 | ND | ND | ND | | 2·4 | 1·4 | 4·3 | |
| **Level 4 –Perinatal factors** | | | | | | | | | | | | |
| Mode of delivery–caesarean section | Vaginal delivery | 6·5 | 11·9 | 10·9 | 1·6 | 0·66 | 4·0 | | 1·6 | 1·1 | 2·5 | |
| Child born prematurely (before week 37) | Not prematurely born | 1·9 | 5·1 | 5·8 | ND | ND | ND | | 2·9 | 1·5 | 5·6 | |
| **Level 5 – Breastfeeding, nutritional status, and previous illness history** | | | | | | | | | | | | |
| Early cessation of exclusive breastfeeding | No early cessation of exclusive breastfeeding | 32·1 | 32·2 | 38·7 | 1·2 | 0·68 | 2·2 | | 1·5 | 1·1 | 1·9 | |
| Not breastfeeding now (or, for cases, just before the diarrhoeal episode started) | Breastfeeding now (or, for cases, just before the diarrhoeal episode started) | 8·4 | 15·3 | 11·1 | 3·6 | 1·5 | 8·6 | | 1·8 | 1·2 | 2·8 | |
| Acute malnutrition | | | | | | | | <0·01[cg] | | | | 0·02[cg] |
| Moderate acute malnutrition (MAM) | No acute malnutrition | 2·2 | 11·9 | 10·4 | 5·9 | 2·2 | 15·8 | | 5·2 | 2·9 | 9·3 | |
| Severe acute malnutrition (SAM) | No acute malnutrition | 0·6 | 6·8 | 2·1 | 9·3[h] | 2·0 | 43·7 | | 4·3 | 1·3 | 14·4 | |
| Acute malnutrition, any (MAM or SAM) | No acute malnutrition | 2·8 | 18·6 | 12·5 | 6·7 | 2·9 | 15·6 | | 5·0 | 2·9 | 8·6 | |
| One or more overnight admissions, since birth | No overnight admissions | 7·3 | 10·2 | 7·9 | 1·0 | 0·40 | 2·6 | | 1·0 | 0·64 | 1·6 | |

(*Continued*)

**Table 2.** (Continued)

| Characteristic | Reference level | Controls[b] (%) (N = 725) | Diarrhoea cases | | Cryptosporidiosis diarrhoea vs controls | | | | Non-cryptosporidiosis diarrhoea vs controls | | | |
|---|---|---|---|---|---|---|---|---|---|---|---|---|
| | | | Cryptosporidiosis[b] (%) (N = 59) | Non-cryptosporidiosis[b] (%) (N = 432) | OR | 95% CI for OR | | Linear trend *P* | OR | 95% CI for OR | | Linear trend *P* |
| | | | | | | from | to | | | from | to | |
| One or more diarrhoea episodes, during the last month | No diarrhoea episodes | 15·0 | 27·1 | 16·9 | 2·4[d] | 1·3 | 4·7 | | 1·2[d] | 0·83 | 1·6 | |
| One or more visits to hospital or health center due to illness, since birth | No visits since birth | 27·2 | 47·5 | 32·9 | 2·4 | 1·4 | 4·1 | | 1·3 | 1·0 | 1·7 | |

OR = odds ratio. CI = confidence interval. WHO = World Health Organization. ND = Not done, due to insufficient number (n < 5) exposed for reliable estimation of OR.

[a] Logistic regression models with the addition of a "base adjustment set" with fixed effect terms for age and gender and random effect intercept terms for enrolment site and season.

[b] Prevalence, after imputing all missing values for exposure variables (see Table A in S1 Appendix for missingness breakdown)

[c] Test for linear trend (P-level), using the ordered categorical variable levels as predictor.

[d] Evidence of selection bias due to missing outcome in some cases (S1 Appendix).

[e] Test for linear trend (P-level), using the continuous variable as predictor.

[f] Models including the sanitation facility variable included a random effect intercept for nurse conducting the interview, due to evidence for differential exposure misclassification for this variable (S1 Appendix).

[g] Test for linear trend also positive (P-level < 0·01) when using MUAC as a continuous predictor variable (in ≥ 6-month-olds).

[h] Few exposed (n = 4) in the SAM subcategory.

intra-level analysis (Table 3). None of these factors were significantly associated with cryptosporidiosis.

Of the level-5 factors, acute malnutrition was strongly associated with both cryptosporidiosis and NCrD in the univariable models, and the strength of the association varied by the degree of malnutrition. Not being breastfed immediately prior to the current diarrhoeal episode was associated with both NCrD and cryptosporidiosis, but early cessation of exclusive breastfeeding (earlier than the WHO minimum recommended age of 6 months) was only significantly associated with NCrD. Previous healthcare attendance was more strongly associated with cryptosporidiosis than with NCrD. After adjusting for other level-5 factors, diarrhoea within the last month was no longer significantly associated with cryptosporidiosis (OR 1.9, 95% CI 0.9 to 3.8; Table 3).

Results from the hierarchical analysis, i.e., taking all hierarchical levels into account, are shown in Table 4. The hierarchical framework determined the order in which variables were included from the intra-level models in the hierarchical analysis, starting with level 1 (base-adjustment-set only), then level 2 (also adjusted for level-1 risk factors), level 3 (also adjusted for level-1 and level-2 risk factors), etc. Most risk factors from the intra-level analyses (Table 3) remained significant in the hierarchical analysis, i.e., after accounting for possible confounding from more distal levels. A notable exception was not being currently breastfed, where the OR for the association with cryptosporidiosis dropped from 3.3 in the intra-level model (Table 3) to 2.0 (95% CI 0.7 to 5.6) after adjustment for more-distal risk factors.

PAF was used as an estimate for the hypothetical relative contribution of each risk factor to the number of cases (Table 4). Some factors were estimated to contribute a small number of cases despite strong risk association. For cryptosporidiosis, the most important contributors

**Table 3. Risk factors for cryptosporidiosis diarrhoea and non-cryptosporidiosis diarrhoea in children under 2-years old, compared with non-diarrhoea controls; odds ratios estimated from separate intra-level multivariable regression models[a].**

| Characteristic | Reference level | Cryptosporidiosis diarrhoea vs controls | | | Non-cryptosporidiosis diarrhoea vs controls | | |
|---|---|---|---|---|---|---|---|
| | | OR | 95% CI for OR | | OR | 95% CI for OR | |
| | | | from | to | | from | to |
| **Level 1 –Socioeconomic factors** | | | | | | | |
| Maternal education | | | | | | | |
| < 1 year | ≥ 8 years | 2·2[b] | 1·0 | 4·7 | | | |
| 1–7 years | ≥ 8 years | 1·1[b] | 0·56 | 2·3 | | | |
| Primary caregiver is not the child's mother | Mother primary caregiver | 5·5 | 1·7 | 17·8 | 3·8 | 1·9 | 7·6 |
| Number of key assets owned by the household ≤ 2 | 3–7 key assets owned | | | | 1·8[c] | 1·3 | 2·4 |
| **Level 2 –Household environmental factors** | | | | | | | |
| Water source | | | | | | | |
| Public tap | Private tap | 3·7[d] | 1·8 | 7·3 | 2·7[e] | 1·9 | 3·9 |
| Surface or rainwater, unprotected well, borehole, or protected spring | Private tap | 0·99[d] | 0·41 | 2·4 | 1·9[e] | 1·3 | 2·8 |
| Water treated by the household (chemicals, boiling, or filtering) before drinking | No water treatment | | | | 1·8 | 1·1 | 2·9 |
| **Level 3 –Hygiene behaviour** | | | | | | | |
| Caregiver will normally wash hands without soap | Normally washes with soap | | | | 2·2 | 1·2 | 3·8 |
| Last stool disposal (from any of the caregiver's children) was unsafe, by the WHO definition | In toilet/latrine, or buried (WHO "safe" stool disposal) | | | | 2·3 | 1·7 | 3·1 |
| **Level 4 –Perinatal factors** | | | | | | | |
| Mode of delivery–caesarean section | Vaginal delivery | | | | 1·6 | 1·0 | 2·4 |
| Child born prematurely (before week 37) | Not born prematurely | | | | 2·8 | 1·4 | 5·5 |
| **Level 5 –Breastfeeding, nutritional status, and previous illness** | | | | | | | |
| Early cessation of exclusive breastfeeding | No early cessation of exclusive breastfeeding | | | | 1·4 | 1·0 | 1·8 |
| Not breastfeeding now (or, for cases, just before the diarrhoeal episode started) | Breastfeeding now (or, for cases, just before the diarrhoeal episode started) | 3·3 | 1·3 | 8·3 | 1·7 | 1·1 | 2·6 |
| Acute malnutrition, any (MAM or SAM) | No acute malnutrition | 6·1 | 2·6 | 14·6 | 4·8 | 2·8 | 8·3 |
| One or more visits to hospital or health center due to illness, since birth | No visits | 2·3 | 1·3 | 4·0 | | | |

Risk factor rows containing empty cells are relevant for either cryptosporidiosis or non-cryptosporidiosis diarrhoea, but not for both.

OR = odds ratio. CI = confidence interval. WHO = World Health Organization. MAM = moderate acute malnutrition. SAM = severe acute malnutrition.

[a] Presented in the table are estimates from multiple regression models with those risk factor variables that remained significant after intra-level modelling; all models were also adjusted for age and gender, with random effect intercepts for enrolment site and season.

[b] Test for linear trend P = 0·04, using the ordered categorical variable levels as predictor.

[c] Test for linear trend P = 0·01, using number of key assets owned as predictor.

[d] Test for linear trend P = 0·98, using the ordered categorical variable levels as predictor.

[e] Test for linear trend P < 0·01, using the ordered categorical variable levels as predictor.

were, from high to low PAF values: public tap water used for drinking, previous illness leading to healthcare attendance, low maternal education, acute malnutrition, and having a primary caregiver other than the mother. By PAF contribution, socioeconomic factors and acute malnutrition were more important for cryptosporidiosis than for NCrD, whereas household environmental and hygiene factors were less important (Table 4).

In the mediation analysis, socioeconomic risk factors for cryptosporidiosis were—to a larger extent than the NCrD risk factors—mediated through proximal risk-factor levels (68% vs 22% of the level-1-PAF, respectively, Table B in S1 Appendix). With this exception, we found only weak evidence for mediation of other distal risk factors.

## Discussion

A key finding from our study was the strong association between acute malnutrition and both cryptosporidiosis and NCrD. For cryptosporidiosis, the association was stronger than has previously been found in studies that compared between diarrhoea cases with and without

**Table 4. Risk factors for cryptosporidiosis diarrhoea and non-cryptosporidiosis diarrhoea in children under 2-years old, compared with non-diarrhoea controls; odds ratios and population attributable fractions estimated from the hierarchical analysis[a].**

| Characteristic | Reference level | Cryptosporidiosis diarrhoea vs controls | | | | Non-cryptosporidiosis diarrhoea vs controls | | | |
|---|---|---|---|---|---|---|---|---|---|
| | | OR | 95% CI for OR | | PAF (%) | OR | 95% CI for OR | | PAF (%) |
| | | | from | to | | | from | to | |
| **Level 1 –Socioeconomic factors** | | | | | | | | | |
| Maternal education | | | | | | | | | |
| < 1 year | ≥ 8 years | 2·2[b] | 1·0 | 4·7 | 19 | | | | |
| 1–7 years | ≥ 8 years | 1·1[b] | 0·56 | 2·3 | NA | | | | |
| Primary caregiver is not the child's mother | Mother primary caregiver | 5·5 | 1·7 | 17·8 | 8 | 3·8 | 1·9 | 7·6 | 4 |
| Number of key assets owned by the household ≤ 2 | 3–7 key assets owned | | | | | 1·8[c] | 1·3 | 2·4 | 11 |
| Socioeconomic factors PAF | | | | | 26 | | | | 14 |
| **Level 2 –Household environmental factors** | | | | | | | | | |
| Water source | | | | | | | | | |
| Public tap | Private tap | 3·8[d] | 1·9 | 7·7 | 27 | 2·5[e] | 1·7 | 3·6 | 15 |
| Surface or rainwater, unprotected well, borehole, or protected spring | Private tap | 1·1[d] | 0·43 | 2·7 | NA | 1·7[e] | 1·2 | 2·5 | 15 |
| Water treated by the household (chemicals, boiling or filtering) before drinking | No water treatment | | | | | 1·8 | 1·1 | 3·0 | 4 |
| Household environmental factors PAF | | | | | 27 | | | | 31 |
| **Level 3 –Hygiene behaviour** | | | | | | | | | |
| Caregiver will normally wash hands without soap | Normally washes with soap | | | | | 1·9 | 1·1 | 3·5 | 3 |
| Child stool disposal unsafe, by the WHO definition | Safe child stool disposal | | | | | 2·3 | 1·6 | 3·1 | 42 |
| Hygiene behaviour factors PAF | | | | | | | | | 44 |
| **Level 4 –Perinatal factors** | | | | | | | | | |
| Mode of delivery–caesarean section | Vaginal delivery | | | | | 1·6 | 1·0 | 2·5 | 4 |
| Child born prematurely (before week 37) | Not born prematurely | | | | | 3·1 | 1·5 | 6·2 | 4 |
| Perinatal factors PAF | | | | | | | | | 8 |
| **Level 5 –Breastfeeding, nutritional status, and previous illness** | | | | | | | | | |
| Early cessation of exclusive breastfeeding | No early cessation of exclusive breastfeeding | | | | | 1·5 | 1·1 | 2·0 | 13 |
| Acute malnutrition, any (MAM or SAM) | No acute malnutrition | 7·2 | 2·9 | 17·8 | 16[f] | 4·6 | 2·6 | 8·0 | 10[f] |
| Moderate acute malnutrition (MAM) | No acute malnutrition | 5·3[e] | 1·8 | 15·5 | 10 | 4·7[g] | 2·5 | 8·7 | 8 |
| Severe acute malnutrition (SAM) | No acute malnutrition | 16·2[eh] | 3·1 | 83·4 | 6[h] | 4·1[g] | 1·2 | 14·6 | 2 |
| One or more visits to hospital or health center due to illness, since birth | No visits | 2·3 | 1·3 | 4·1 | 27 | | | | |
| Breastfeeding, nutritional status and previous illness factors PAF | | | | | 39 | | | | 22 |
| **Summary PAF for all levels** | | | | | **67** | | | | **76** |

Risk factor rows containing empty cells are relevant for either cryptosporidiosis or non-cryptosporidiosis diarrhoea, but not for both.

OR = odds ratio. CI = confidence interval. PAF = Population attributable fraction. WHO = World Health Organization. MAM = moderate acute malnutrition.

SAM = severe acute malnutrition. NA = Not applicable, i.e., PAF not estimated as this subcategory was not a significant risk factor for cryptosporidiosis.

[a] All multiple-regression models adjusted for age and gender, with random effect intercepts for enrolment site and season.

[b] Test for linear trend P = 0·04, using the ordered categorical variable levels as predictor.

[c] Test for linear trend P = 0·01, using number of key assets owned as predictor.

[d] Test for linear trend P = 0·84, using the ordered categorical variable levels as predictor.

[e] Test for linear trend P < 0·01, using the ordered categorical variable levels as predictor.

[f] PAF estimates for acute malnutrition need to be interpreted with caution due to the cross-sectional evaluation of this exposure-outcome association (see Discussion).

[g] Test for linear trend P = 0·03, using the ordered categorical variable levels as predictor.

[h] Few exposed (n = 4) in the SAM subcategory.

cryptosporidiosis [12–19], and may be representative of a general healthcare-presenting population with diarrhoea, i.e., not restricted to children with moderate-to-severe or acute diarrhoeal episodes. MAM was less strongly associated with cryptosporidiosis than SAM but was far more common. The WHO provides no specific guidance for the management of diarrhoea in children with MAM, besides the promotion of supplementary foods [29, 35]. Regrettably, most Ethiopian children with MAM, including in our study area, are excluded from supplementary feeding programmes, as they reside within areas that are not classified as food insecure [36]. Case-finding and proper nutritional rehabilitation for all children with acute malnutrition should be a priority. The only currently approved therapeutic for cryptosporidiosis, nitazoxanide, is still not widely used in Africa, and is not registered in Ethiopia. The drug is assumed to have a modest effect on cryptosporidiosis in HIV-negative children with acute malnutrition. However, it is worth noting that the only randomized controlled trial that enrolled HIV-seronegative children with acute malnutrition, included only 11 severely wasted and 6 moderately wasted participants, all 1-3-years old, yet demonstrated a significant effect on both diarrhoeal duration and case fatality [37]. Nitazoxanide treatment for cryptosporidiosis in children with MAM and SAM could be explored as a simple intervention, if facilitated by low-tech point-of-care tests with proven accuracy [26]. We agree with calls for further research and development of new and more effective drugs, while simultaneously recognizing the potential benefits of wider global use of nitazoxanide [38–40].

Previous healthcare visits were common in all children who presented with diarrhoea, particularly in cryptosporidiosis, where this association remained strong also after controlling for confounding from more distal levels. Previous history of illness and acute malnutrition can be considered as criteria to help prioritize which children to test for cryptosporidiosis, should testing capacity be limited. Caregivers of this higher-risk group of children might be cost-effective targets for secondary prevention of cryptosporidiosis (e.g., intervention bundles containing tools and advice on how to manage diarrhoea and malnutrition), and incentives and advice regarding when to return for review and diagnostic testing (e.g., explaining the benefits of treatment, skipping of queues for reassessment, and transport reimbursements).

Unlike previous puzzling reports [8, 9, 11], not having piped water access in the household was a risk factor for both cryptosporidiosis and NCrD, similar to what has been found for all-cause diarrhoea [41]. It is interesting that public tap water, considered an improved source, was more strongly associated with cryptosporidiosis than consumption of water from unimproved sources. This might reflect poor water quality, i.e., faecal contamination of the piped water at the point of supply or collection. Alternatively, it could be a marker for insufficient water quantity affecting household hygiene in a way that is not fully captured by the level-3-variables. The finding warrants examining the piped water at various supply points, and maybe the pump handles themselves, for both faecal indicator bacteria and *Cryptosporidium* oocysts.

Ethiopia is one of the poorest countries in the world, yet, over the last two decades, has demonstrated an impressive reduction in extreme poverty, while maintaining relative wealth equality [42]. Nevertheless, we found that socioeconomic status, particularly when assessed by maternal education, was a large contributor to healthcare-attended cryptosporidiosis in our study area. It might be possible to target caregivers with specific interventions related to hygiene behaviour and nutrition, but it seems wise to first obtain local data on the extent to which care-related risks act through behaviours and beliefs that are amenable to intervention. In our study population, the proposed effect appears to be largely mediated through intermediate risk factors (Table B in S1 Appendix). There is an important methodological lesson implicit in this finding: had we performed multivariable analysis without the application of a causal framework, socioeconomic status (specifically, low maternal education) would have appeared

to be unimportant. For NCrD, the effect of such overadjustment bias would be less dramatic, but would have resulted in lower estimates of both the strength of association and contribution to case load from level-1-factors.

Several limitations need to be considered when interpreting our findings. Defining acute malnutrition in the context of diarrhoea is challenging, as weight is affected by dehydration. To address this, MUAC was used to classify acute malnutrition in ≥ 6-month-olds, as it is less vulnerable to dehydration than weight [43, 44]. Also, the conceptual scheme used in this analysis is a simplification of the complex relationship between malnutrition and diarrhoea. It is difficult to disentangle in individual children whether malnutrition causally increases the risk of diarrhoea, whether diarrhoea inflicts a nutritional insult, or both. While most researchers accept that episodes of acute malnutrition increase the risk of all-cause diarrhoea [45], the evidence for the degree to which pre-existing malnutrition increases the risk of specific diarrhoeal syndromes, including cryptosporidiosis, is sparse (S1 Appendix) [46]. Some birth-cohort studies indicate that cryptosporidiosis can cause reduced ponderal growth in the 6-month period after an episode [47, 48] and a recent publication reporting on a large multi-country study of infants and toddlers with moderate-to-severe diarrhoea identified *Cryptosporidium* infection as a predictor of linear growth faltering [49]. In the current study, we were unable to disentangle the proportion of the observed association that was due to acute malnutrition (MAM or SAM) increasing the risk of cryptosporidiosis, and how much of the association was due to the current cryptosporidiosis episode leading to acute malnutrition. However, because most diarrhoeal episodes were short, we carefully speculate that the former link may have outweighed the latter (Table C in S1 Appendix). While we believe the OR describing the association is valid, the PAF for acute malnutrition could be an overestimate, for both cryptosporidiosis and NCrD. Appropriate analysis of case-control studies, where children with and without acute malnutrition are followed up for pathogen-attributed diarrhoeal episodes [50], will be necessary to explore this important question further.

Second, the apparent lack of association between early cessation of exclusive breastfeeding and cryptosporidiosis should be interpreted with particular care, as there were only three cases in the 0-5-month group, where introduction of foods and liquids other than breastmilk is likely to pose the highest risk (Table 1). Birth cohorts report a peak in cryptosporidiosis incidence at 6–11 months of age [9, 51], which is consistent with community-based studies that indicate a significant protective effect of breastfeeding [8, 52]. Not breastfeeding currently was significantly associated with cryptosporidiosis and NCrD in the initial models, but not in the hierarchical analysis. A possible explanation is that breastfeeding is, at least in part, a marker for one or more of the distal-level risk factors. Confounding from socioeconomic factors can bias inference about all exposure-outcome relationships at more proximal levels and is considered particularly important for breastfeeding [53].

Third, the counterintuitive finding of an association between point-of-use water treatment and NCrD should be interpreted with caution, as water treatment was uncommon overall. It is possible that the observed association is confounded by some aspect of water quality not fully captured by our water source variable.

Fourth, at least a third of the total cryptosporidiosis case load could not be attributed to any of the identified risk factors (summary PAF for all levels, 67%, Table 4). Our list of investigated risk factors was far from exhaustive. Some variables were omitted due to high risk of selection bias (e.g., vaccination status, diarrhoea in close contacts) [19], or were omitted by design (e.g., variables related to food handling, floor covering, day-care attendance, swimming).

Fifth, we did not adjust for all factors that may influence healthcare seeking, such as disease severity or dehydration, as healthcare presentation was integral to our outcome definition (Fig 1), and adjustment could therefore have induced a bias. Likewise, we are not able to quantify

how caretakers' healthcare-seeking decisions may have impacted the observed association between acute malnutrition and cryptosporidiosis.

Sixth, due to the difference in the number of children between the case sets, limited by the parent study (S1 Appendix), and because this is a case-case-control rather than a case-case study, strong inferences cannot be made based on the observed differences in OR between the cryptosporidiosis and NCrD groups. Most of the confidence intervals obtained from the analysis are sufficiently narrow to allow meaningful interpretation of their corresponding point estimates. However, we note that for some putative risk factors the confidence intervals for the OR are wide and therefore difficult to interpret, reflecting uncertainty due to the limited number of children and events for some of the comparisons, e.g., in the multivariable cryptosporidiosis vs control comparisons.

Finally, the evidence presented here can at best be used as support for a causal relationship between the identified risk factors and cryptosporidiosis, as there have been few interventional studies. For diarrhoea in general, there is now a large body of both observational and interventional evidence to support a causal role of underprivileged access to water/sanitation/hygiene, perinatal factors, lack of breastfeeding, and malnutrition [7]. There are important similarities in transmission (e.g., faecal-oral) and host susceptibility (e.g., nutrition) between various diarrhoeal infections, which likely also explain that there were many common risk factors for cryptosporidiosis and NCrD. Nevertheless, by using a hierarchical case-case-control analysis, we do observe some differences that may plausibly be related to characteristics of the *Cryptosporidium* parasite. These include stronger association with piped water from public, rather than private, taps, which could be related to the environmental robustness of *Cryptosporidium* oocysts, and higher risk associated with previous healthcare attendance and acute malnutrition, which could be related to a particular need for a healthy immune response to prevent illness and resolve symptoms. An important next step in exploring this puzzle will be obtaining more evidence from preventive interventions against diarrhoea where outcomes are considered by aetiology [54], and trials that examine the role of pharmacological treatment of cryptosporidiosis as part of the nutritional rehabilitation of malnourished children.

## Supporting information

**S1 Checklist. STROBE checklist.**
(PDF)

**S1 Appendix. Supplementary appendix.** Table A in S1 Appendix. Distribution of case and control subjects according to all exposures, with counts and proportions of missing values. Table B in S1 Appendix. Hierarchical mediation analysis of risk factors for cryptosporidiosis and non-cryptosporidiosis diarrhoea, in children under 2 years old. Table C in S1 Appendix. Duration of diarrhoea, on enrolment, in cryptosporidiosis and non-cryptosporidiosis diarrhoea cases, in children under 2 years old.
(PDF)

**S1 Dataset. Anonymized dataset.**
(XLS)

## Acknowledgments

The authors thank Wakjira Kebede and Yonas Alemu who performed IFAT during parts of the study; Ola Bjørang who performed *Cryptosporidium* qPCR; all other Jimma University academic staff and clinical and laboratory staff in JMC and SHC who were involved in the study; laboratory staff in Vestfold Hospital Trust for their help with implementing the study; all

children who participated in the study, and the children's caregivers. We received *Cryptosporidium* ELISA kits free of charge from TECHLAB (Blacksburg, VA, USA).

## Author Contributions

**Conceptualization:** Øystein Haarklau Johansen, Alemseged Abdissa, Mike Zangenberg, Nina Langeland, Lucy J. Robertson, Kurt Hanevik.

**Data curation:** Øystein Haarklau Johansen, Alemseged Abdissa, Mike Zangenberg.

**Formal analysis:** Øystein Haarklau Johansen, Mike Zangenberg, Kurt Hanevik.

**Funding acquisition:** Øystein Haarklau Johansen, Mike Zangenberg, Nina Langeland, Lucy J. Robertson, Kurt Hanevik.

**Investigation:** Øystein Haarklau Johansen, Alemseged Abdissa, Mike Zangenberg, Beza Eshetu, Bizuwarek Sharew, Halvor Sommerfelt, Nina Langeland, Lucy J. Robertson, Kurt Hanevik.

**Methodology:** Øystein Haarklau Johansen, Alemseged Abdissa, Mike Zangenberg, Zeleke Mekonnen, Beza Eshetu, Sabrina Moyo, Halvor Sommerfelt, Nina Langeland, Lucy J. Robertson, Kurt Hanevik.

**Project administration:** Øystein Haarklau Johansen, Alemseged Abdissa, Nina Langeland, Kurt Hanevik.

**Resources:** Øystein Haarklau Johansen, Alemseged Abdissa, Mike Zangenberg, Zeleke Mekonnen, Bizuwarek Sharew, Lucy J. Robertson, Kurt Hanevik.

**Software:** Øystein Haarklau Johansen, Alemseged Abdissa, Mike Zangenberg.

**Supervision:** Øystein Haarklau Johansen, Alemseged Abdissa, Mike Zangenberg, Zeleke Mekonnen, Beza Eshetu, Bizuwarek Sharew, Nina Langeland, Lucy J. Robertson, Kurt Hanevik.

**Validation:** Øystein Haarklau Johansen, Alemseged Abdissa, Mike Zangenberg, Nina Langeland, Kurt Hanevik.

**Visualization:** Øystein Haarklau Johansen.

**Writing – original draft:** Øystein Haarklau Johansen.

**Writing – review & editing:** Øystein Haarklau Johansen, Alemseged Abdissa, Mike Zangenberg, Zeleke Mekonnen, Beza Eshetu, Bizuwarek Sharew, Sabrina Moyo, Halvor Sommerfelt, Nina Langeland, Lucy J. Robertson, Kurt Hanevik.

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
