## [Decision Letter · Decision Letter 0]

12 Dec 2021

Dear Dr. Johansen,

Thank you very much for submitting your manuscript "A comparison of risk factors for cryptosporidiosis and non-cryptosporidiosis diarrhoea: a hierarchical case-case-control study in Ethiopian children" for consideration at PLOS Neglected Tropical Diseases. 

As with all papers reviewed by the journal, your manuscript was reviewed by members of the editorial board and by several independent reviewers. In light of the reviews (below this email), we would like to invite the resubmission of a significantly-revised version that takes into account the reviewers' comments. 

We cannot make any decision about publication until we have seen the revised manuscript and your response to the reviewers' comments. Your revised manuscript is also likely to be sent to reviewers for further evaluation.

Sincerely,

Sitara SR Ajjampur

Associate Editor

Pikka Jokelainen

Deputy Editor

Reviewer's Responses to Questions

**Key Review Criteria Required for Acceptance?**

**Methods**

-Are the objectives of the study clearly articulated with a clear testable hypothesis stated?

-Is the study design appropriate to address the stated objectives?

-Is the population clearly described and appropriate for the hypothesis being tested?

-Is the sample size sufficient to ensure adequate power to address the hypothesis being tested?

-Were correct statistical analysis used to support conclusions?

-Are there concerns about ethical or regulatory requirements being met?

Reviewer #1: (No Response)

Reviewer #2: The objectives of study are clearly described. Study design is appropriate to address the objectives

Basic exposures and outcome variables need to define in summary and should include in the manuscript for clear understanding of the readers. 

The samples size was not discussed. The samples size is unlikely sufficient to address all the risk factors and multivariate analysis

Statistical analysis was good

The study fulfilled the criteria of ethical or regulatory requirement

**Results**

-Does the analysis presented match the analysis plan?

-Are the results clearly and completely presented?

-Are the figures (Tables, Images) of sufficient quality for clarity?

Reviewer #1: (No Response)

Reviewer #2: The analysis was clearly described

The results presented well 

Tables and figures are sufficient quality for clarity

**Conclusions**

-Are the conclusions supported by the data presented?

-Are the limitations of analysis clearly described?

-Do the authors discuss how these data can be helpful to advance our understanding of the topic under study?

-Is public health relevance addressed?

Reviewer #1: (No Response)

Reviewer #2: The conclusion is OK.

There are some limitations which need to address.

The authors addressed the data clearly.

The manuscript has public health importance. The study findings are important for researchers and policy makers.

**Editorial and Data Presentation Modifications?**

Reviewer #1: (No Response)

Reviewer #2: The manuscript needs minor revision.

**Summary and General Comments**

Reviewer #1: (No Response)

Reviewer #2: The manuscript compares the risk factors of cryptosporidiosis and non- cryptosporidiosis diarrhea cases and cryptosporidiosis vs. control without diarrhea.

The manuscript is well written and addressed case-case-control study design in children < 2 years who suffers most with cryptosporidium diarrhea and infection. This has public health importance to understand risk factors of cryptosporidiosis and to take preventive actions. 

However, basic exposures and outcome variables need to define in summary and should include in the manuscript for clear understanding of the readers. 

Few following comments are recommend to include in the manuscript preferably or to address. 

Define diarrhea case and cryptosporidiosis in the manuscript. Risk factors for diarrhoea cases recruited in community might differ for diarrhoea cases recruited in health care settings. Similarly, diarrhoea severity might have impact on cryptosporidiosis and risk factors. However, this has not been addressed in the manuscript 

Moderate acute malnutrition was important predictor for cryptosporidiosis. How SAM/MAM was defined? Diarrhea and dehydration have impact on weight and to assess malnutrition status. This might overestimate MAM/SAM. This should include in discussion. 

What was the quantitative cut off value for diagnostic accuracy of cryptosporidiosis? How presence of other pathogens addressed in this quantitative cut off value for diagnosis of cryptosporidiosis?

How sufficient power was maintained to address risk factor analysis?

Minor comments:

Page 2, L-50: The authors recommend following up children with previous illness. The sentence should be more conservative. This might not be feasible in low resources settings. In multivariate analysis, it is indicated as health centre visit due to illness since birth. The diarrhoea was not shown as risk factor in this analysis.

P-22, l-293: The previous illness as mentioned above.

Page 23, L 316-317. Specify age and HIV status to indicate modest effect.

PLOS authors have the option to publish the peer review history of their article (what does this mean?). If published, this will include your full peer review and any attached files.

Reviewer #1: Yes: Emily L Deichsel

Reviewer #2: Yes: M. Jahangir Hossain
---

## [Decision Letter · Decision Letter 1]

18 Apr 2022

Dear Dr. Johansen,

Thank you very much for submitting your manuscript "A comparison of risk factors for cryptosporidiosis and non-cryptosporidiosis diarrhoea: a case-case-control study in Ethiopian children" for consideration at PLOS Neglected Tropical Diseases. As with all papers reviewed by the journal, your manuscript was reviewed by members of the editorial board and by several independent reviewers. The reviewers appreciated the attention to an important topic. Based on the reviews, we are likely to accept this manuscript for publication, providing that you modify the manuscript according to the review recommendations. 

Sincerely,

Sitara SR Ajjampur

Associate Editor

Pikka Jokelainen

Deputy Editor

Reviewer's Responses to Questions

**Key Review Criteria Required for Acceptance?**

**Methods**

-Are the objectives of the study clearly articulated with a clear testable hypothesis stated?

-Is the study design appropriate to address the stated objectives?

-Is the population clearly described and appropriate for the hypothesis being tested?

-Is the sample size sufficient to ensure adequate power to address the hypothesis being tested?

-Were correct statistical analysis used to support conclusions?

-Are there concerns about ethical or regulatory requirements being met?

Reviewer #1: The methods well written and appropriate. A few suggestions. 

Methods state that you applied quantitative cut offs to distinguish asymptomatic and symptomatic crytpo. It’s my understanding that you used quantitative results along with other test results to diagnose crypto infection. Those above the cutoff threshold still had diarrhea (symptoms), just of another cause. These are not asymptomatic crypto infections. That would be the detection of crypto in the controls. 

Please also clarify in the main text that no stool testing was done on controls. 

How did you handle age for exclusively breastfed for shorter than 6 months. This should be limited to only children who are at least 6 months of age. Based on this discussion it seems those under 6 months were grouped into the not breastfed for 6 months group. 

I suggest adding a bit more detail about the PAF mediation analysis completed in the main paper so the reader does not have to see the appendix for this method.

Reviewer #2: (No Response)

**Results**

-Does the analysis presented match the analysis plan?

-Are the results clearly and completely presented?

-Are the figures (Tables, Images) of sufficient quality for clarity?

Reviewer #1: Results are presented well and presented clearly.

Reviewer #2: (No Response)

**Conclusions**

-Are the conclusions supported by the data presented?

-Are the limitations of analysis clearly described?

-Do the authors discuss how these data can be helpful to advance our understanding of the topic under study?

-Is public health relevance addressed?

Reviewer #1: I think the authors overstate the strong association between crypto diarrhea and malnutrition. To some extent, this association represents the association between any diarrhea and care-seeking and malnutrition (as demonstrated by the NCrD cases). It is established that improving nutrition reduces infectious disease morbidity and the severity of that morbidity. I don’t think you can claim malnutrition and unsafe water as unique characteristics of crypto infection based on this analysis.

Reviewer #2: (No Response)

**Editorial and Data Presentation Modifications?**

Reviewer #1: (No Response)

Reviewer #2: (No Response)

**Summary and General Comments**

Reviewer #1: The author’s present a thoughtful, well written, and thorough analysis of risk factors for cryptosporidium and diarrhea in a population of children under two in Ethiopia. The manuscript applies a novel analytic strategy to covers an important and relevant public health topic and is much improved.

Reviewer #2: The Authors have sufficiently addressed earlier comments. The manuscript could be accepted after addressing few following comments.

Abstract, Background, L-34: The authors mentioned in the background that there is a "broad overlap in risk factors between cryptosporidiosis and other diarrhoeal aetiologies". However, this is not very clear how the authors addressed this issue in the manuscript. 

Abstract, Methodology/Principal findings: Background, L-40: The authors mentioned that "We applied quantitative cut-offs to distinguish between asymptomatic and symptomatic Cryptosporidium infection". Many of controls were positive to Cryptosporidium infection. However, I don’t see to address this in laboratory analysis section. This sentence is not consistent with the contents of manuscript in method section. 

Abstract, Methodology/Principal findings: Background, L: 43-45: The statement “Side-by-side comparisons indicate that socioeconomic factors and public tap water use were more strongly associated with cryptosporidiosis than with non-cryptosporidiosis diarrhoea” is likely not consistent as per table 3 and table 4 especially for socioeconomic factors.

Selection of cases and controls, page 5, L: 122-123: The definition of diarrhea or dysentery needs to be specific. Was the diarrhoea acute or chronic? If diarrhoea is acute, how was it defined to differentiate from chronic diarrhoea?

 Fig 1, Level 5: Minor editions: “Previous illness” is termed as “previous healthcare attendance” in revised manuscript. It is recommended to make it consistent throughout the manuscript

Table 1, page 11: The percentages have not corrected/updated for age in months in controls, Cryptosporidiosis and non-cryptosporidiosis. 

Table A in S1 Appendix: The number of animal owned (n=1572) in Tab A in S1 Appendix is confusing. The number is more than the denominator (n=725). Are the authors indicating households owned animals or the total number of animals all the household of study participants? 

Minor editions: 

Introduction, page 4, Line 78: Add “.” at the end of the sentence. 

Introduction, page 4, L-96: Delete “-“

PLOS authors have the option to publish the peer review history of their article (what does this mean?). If published, this will include your full peer review and any attached files.

Reviewer #1: Yes: Emily L Deichsel

Reviewer #2: No

Figure Files:

Data Requirements:

Reproducibility:

References

---

## [Editor Report · Decision Letter 2]

17 May 2022

Dear Dr. Johansen,

We are pleased to inform you that your manuscript 'A comparison of risk factors for cryptosporidiosis and non-cryptosporidiosis diarrhoea: a case-case-control study in Ethiopian children' has been provisionally accepted for publication in PLOS Neglected Tropical Diseases.

Best regards,

Sitara SR Ajjampur

Associate Editor

Pikka Jokelainen

Deputy Editor

---

## [Editor Report · Acceptance letter]

31 May 2022

Dear Dr. Johansen,

We are delighted to inform you that your manuscript, "A comparison of risk factors for cryptosporidiosis and non-cryptosporidiosis diarrhoea: a case-case-control study in Ethiopian children," has been formally accepted for publication in PLOS Neglected Tropical Diseases.

Best regards,

Shaden Kamhawi

co-Editor-in-Chief

Paul Brindley

co-Editor-in-Chief
